# Synthesis and Antibacterial Activity of (AgCl, Ag)NPs/Diatomite Hybrid Composite

**DOI:** 10.3390/ma13153409

**Published:** 2020-08-02

**Authors:** Zhanar Kubasheva, Myroslav Sprynskyy, Viorica Railean-Plugaru, Paweł Pomastowski, Aliya Ospanova, Bogusław Buszewski

**Affiliations:** 1Faculty of Chemistry and Chemical Technology, Al-Farabi Kazakh National University, Almaty 050040, Kazakhstan; bekisanova@gmail.com (Z.K.); ospanova_a@mail.ru (A.O.); 2Department of Environmental Chemistry and Bioanalytics, Faculty of Chemistry, Nicolaus Copernicus University in Torun, 87-100 Torun, Poland; bbusz@umk.pl; 3Interdisciplinary Center for Modern Technologies, Nicolaus Copernicus University in Torun, 87-100 Torun, Poland; rviorela@yahoo.com (V.R.-P.); pawel_pomastowski@wp.pl (P.P.)

**Keywords:** diatomite, silver nanoparticles, (AgCl, Ag)NPs/diatomite composite, antibacterial activity

## Abstract

In the present research, hybrid (AgCl, Ag)NPs/diatomite composites were synthesized by direct impregnation with aqueous silver nitrate solutions. The silver chloride nanoparticles (AgCl-NPs) were formed as an effect of the exchange reaction when silver interacted with the diatomite mineral impurity halite. Nanoparticles of metallic silver (AgNPs) were created by the reduction of silver ions under the influence of hydrogen peroxide. The content of silver chloride nanoparticles in the (AgCl, Ag)NPs/diatomite composite was limited by the content of the halite in the used diatomite. Samples of natural diatomite and synthesized (AgCl, Ag)NPs/diatomite composites were examined by using scanning electron microscopy, transmission electron microscopy, X-ray powder diffraction, infrared spectroscopy and thermogravimetric analysis. Moreover, the antibacterial potential of synthesized composites was also studied using the MIC (minimal inhibitory concentration) method against the most common drug-resistant microorganisms in the medical field: Gram-positive *Staphylococcus aureus* and Gram-negative *Klebsiella pneumoniae*. The obtained hybrid (AgCl, AgNPs)/diatomite composites were shown to have antimicrobial potential. However, widespread use requires further study by using various microorganisms and additional cytotoxic studies on eukaryotic systems, e.g., cell lines and animal models.

## 1. Introduction

Recently, many natural mineral matrices have been actively used as support substrates for metal nanoparticles, e.g., silver nanoparticles, in the production of nanocomposites with specific chemical-biological properties [1]. The interest in mineral matrices appears from the fact that in the case of the conventional synthesis of metal nanoparticles in the form of colloidal solutions, problems such as instability of nanoparticles and their subsequent agglomeration are revealed [2,3,4]. Furthermore, natural minerals as carriers of metal nanoparticles are nontoxic and environmentally friendly.

The necessary conditions for the synthesis of silver nanoparticle composites based on mineral substrates such as quartz [5,6,7,8], talc [9], diatomite [2,10,11], kaolinite [12], montmorillonite [13,14,15,16], zeolite [17,18,19] and mesoporous silica [20,21] have been established. The formation of silver nanoparticles on the surface of mineral substrates can be obtained by ion exchange in silver nitrate solution and further reduction by different methods, such as thermal treatment [16], UV-irradiation, which allows the regulation of the size of nanoparticles by manipulating irradiation parameters [22], methods of hydrothermal synthesis [19] and sonochemical reduction [14,20], as well as by using wet reduction methods [23,24,25,26,27,28,29,30,31,32]. Wet reduction methods characterized by the formation of nanoparticles of silver on surfaces of mineral matrices involve mineral matrices being impregnated with an aqueous solution of AgNO_3_ and the subsequent reduction of silver ions to metallic silver. As a reducing agent, various compounds such as D-glucose [23], ascorbic acid [24], sodium acrylate [25], sodium borohydride [12,13,14,15,26,27,28], sodium citrate [20,28], aspartic acid [29], tanning acid [30], poly(vinylpyrrolidone), carboxymethyl cellulose and gelatin [31] are used. It has also been reported that the formation of silver nanoparticles could be carried out by a reduction in ethanol solution in the presence of certain surfactants [32], as well as with hydrazine dihydrochloride in a slightly alkaline solution [33] or in an acidic medium with sodium formaldehyde sulfoxylate [34]. Sodium borohydride remains the most commonly used reducing agent. Layered clay minerals were used to host silver nanoparticles in the interlayer space and to enhance their bond to the carrier surface. Pre-intercalation of the interlayer space by high molecular weight compounds, such as polyacrylonitrile [15] and dimethylsulfoxide [12], was carried out. Graphene oxide coating was used to improve AgNP stability in the diatomite composite [10].

The use of mineral matrices as carriers of silver nanoparticles can be promising from the environmental and economical points of view. The practical applications of such composites may be various. Silica-based composites are used as an adsorbent for purifying water from mercury ions [6], in catalytic systems [7] and as antibacterial biofilms [8]. Montmorillonite-based nanocomposites (Ag/MMT) were applied in biopolymer matrices for packaging materials [16]. Diatomite coated with silver nanoparticles is used for antimicrobial water treatment [11,35]. Silver-containing composites based on zeolites [17,18] and diatomite [10,11] showed active antibacterial properties. 

Therefore, the main purpose of this study was, firstly, to obtain a hybrid composite (AgCl, Ag)NPs/diatomite) based on natural diatomite and silver nanoparticles in the form of metallic silver (AgNPs) and silver chlorides (AgCl-NPs) simultaneously and, secondly, to investigate the antibacterial potential of obtained composites against Gram-positive (*Staphylococcus aureus*) and Gram-negative (*Klebsiella pneumoniae*) bacteria. In the present experiment, the chloride ion source was the mineral impurity halite, naturally present in the diatomite support. From an application perspective, the obtained formulations can be found as bioactive and biocompatible materials in the medicine, cosmetic or food industry (packaging materials).

## 2. Materials and Methods

### 2.1. The Diatomite Sample

The source of natural diatomite was the Mugodzhar deposit (Kazakhstan, Aktobe region). The diatomite samples were prepared by hand-crushing in a porcelain mortar with subsequent sieving to obtain a granulometric fraction with particle size < 0.30 mm.

### 2.2. Characterization Methods and Instrumentation

The elemental composition of natural diatomite and the sizes and distribution of nanoparticles in synthesized (AgCl, Ag)NPs/diatomite composite structures were determined by using scanning electron microscopy (SEM, LEO 1430 VP, Leo Electron Microscopy Ltd., Cambridge, United Kingdom) coupled with an energy dispersive X-ray (EDX) detector (XFlash 4010, Bruker AXS, Bremen, Germany) and transmission electron microscopy (TEM, FEI Tecnai F20 X-Twintool, FEI Europe, Frankfurt/Main, Germany). The X-ray powder diffraction (XRD) method was used to determine the mineral composition of natural diatomite, and the formation of Ag nanoparticles was examined using a Philips X ‘Pert Pro diffractometer (XRD, Malvern Pananaliytical Ltd., Malvern, United Kingdom) with Cu-Kα-radiation (γ = 0.1541 nm, 40 kV, 30 mA). The XRD pattern data were collected over an angular range of 5–100°2θ with step sizes of 0.01. Functional groups of diatomite samples were determined on an FTIR spectrometer (FTIR ATR, Vertex 70, Bruker Optics, Bremen, Germany) equipped with a DLatTGS detector. The FTIR spectra were recorded by averaging 64 scans in the wavenumber range of 400 cm^−1^ to 4000 cm^−1^ with a resolution of 4 cm^−1^. Thermogravimetric analysis was performed on a TGA-DSC analyzer (Thermal Analysis Instruments, SDT 2960, TA Instruments, Warszawa, Poland) in the temperature range of 20–1100 °C. The analysis was carried out at a linear heating rate of 10 °C/min in a nitrogen gas stream [36].

### 2.3. Preparation of (AgCl,Ag)NPs/Diatomite Composite 

Synthesis of (AgCl, Ag)NPs/diatomite composites was carried out by the impregnation of diatomite with aqueous silver nitrate solution. An initial solution of AgNO_3_ (POCH, Gliwice, Poland) with a concentration of 1000 mg/L was prepared. A natural diatomite mass of 1.0 g was added to a silver nitrate solution (100 mL) with the following Ag ion concentrations: 100 mg/L, 500 mg/L and 1000 mg/L expressed as 1, 5 and 10% in calculations of 1, 5 and 10% of silver relative to the mass of used diatomite. The obtained suspensions were stirred for 30 min and then basified with 0.1 M NaOH to pH = 9. 

During the interaction of the silver nitrate solution with diatomite, according to the inorganic chemistry of silver [37], we assumed that the silver chloride nanoparticles (AgCl-NPs) were formed immediately as a result of the exchange reaction of silver nitrate with the halite mineral impurity (NaCl) in the diatomite:AgNO_3_(aq) + NaCl(s) = NaNO_3_(aq) + AgCl(s)(1)

Then, taking into consideration the fact that the silver nitrate content in the prepared slurry was in excess compared to the halite mineral impurity, the silver ion residues Ag in the alkaline medium were oxidized to silver oxides after completion of the exchange reaction of the silver nitrate with the halite mineral impurity:2AgNO_3_(aq) + 2NaOH(aq) = 2NaNO_3_(aq) +Ag_2_O(s) + H_2_O(2)

Further reduction of the formed silver oxide was carried out with hydrogen peroxide. The reducing agent was added in a 1:3 molar ratio of AgNO_3_/H_2_O_2_. The suspension was stirred for 15 min at 300 rpm until the complete reduction of the silver ions:Ag_2_O(s) + H_2_O_2_(aq) = 2Ag^0^(s) + H_2_O(aq) + O_2_(g)(3)

The addition of the reducing agent changed the pH of the solution from 9.02 to 6.88. Next, the (AgCl, Ag)NPs/diatomite composite was washed five times with deionized water, centrifugated (Centrifuge 9000, MPW-251 rpm) and dried at 110 °C. 

### 2.4. Antimicrobial Assay of (AgCl, Ag)NPs/Diatomite Composite

The antimicrobial properties of the obtained (AgCl, Ag)NPs/diatomite composites (0.71% Ag/diatomite, 4.65% Ag/diatomite, 7.21% Ag/diatomite) were investigated by a minimum inhibitory concentration (MIC) assay. The method was performed using the Miller Hilton (MH) broth medium according to Clinical and Laboratory Standards Institute (CLSI) procedures (with suitable changes) and the resazurin-based 96-well plate microdilution method [38]. For this purpose, two different bacterial strains were used: one Gram (+)—*Staphylococcus aureus* strain and one Gram (−)—*Klebsiella pneumoniae* strain. Firstly, the bacterial cells were inoculated in MH broth media for 24 h at 37 °C. Then, in 96-well flat-bottom plates (Sigma Aldrich, Poznan, Poland), the cultured bacterial strain (1 × 10^6^ CFU/mL) and different concentrations (10 mg/mL, 5 mg/mL, 2.5 mg/mL, 1.25 mg/mL, 0.625 mg/mL, 0.312 mg/mL, 0.156 mg/mL) of the investigated formulations were mixed in a ratio of 1:1. Subsequently, to each well, 12 μL of dye from a resazurin-based in vitro toxicology assay kit (Sigma-Aldrich, St. Louis, MO, USA) was added. Once the samples had been prepared, plates were kept at 37 °C under continuous stirring for 24 h. Natural diatomite (raw material) served as a control. The MIC value was determined by changes in the indicator color from blue to pink. All the experiments have been performed in triplicate.

## 3. Result and Discussion

### 3.1. Characteristic of Natural Diatomite

The elemental composition of the natural diatomite obtained by the SEM-EDX (Figure 1A) method shows that silicon, oxygen, aluminum and iron are the main elements, while sodium, magnesium and chlorine are the minor elements. The structure and morphology of natural diatomite were studied by SEM (Figure 1B). The microstructure of the sample mainly consists of numerous frustules of diatomaceous algae with pores in the range of 0.8–1 μm. Fragments of non-uniformly distributed particles of laminated plates are also found. This phenomenon is related to the content of the clay minerals in the mineral composition of diatomite rock. The results of the XRD analysis (Figure 1C) reveal that the diatomite used is composed of such minerals as kaolinite, illite, halite, quartz and amorphous silica, reflected by the following peaks at 2θ: 12.43°, 25.03° for kaolinite, 17.69°, 21.02°, 35.10° for illite, 31.69°, 45.45° for halite and 21.02°, 26.79°, 50.25° for quartz. The presence of amorphous silica of diatom frustules is indicated by a broad peak in the range of 2θ between 20° and 35°. Functional groups, determined by the FTIR-ATR method, in natural diatom are represented in Figure 1D. The main spectral peaks of functional groups are detected at 3697, 3621, 1631, 1026, 913, 796, 694, 525 and 451 cm^−1^. The signals generated at 3694 and 3621 cm^−1^ correspond to tension vibrations of Al-OH and Si-OH groups [39,40]. The strip appearing at 1631 cm^−1^ is related to the vibration of H-O-H due to the stretching and bending of adsorbed water on the surface of silica [41]. The intensity of the adsorption strip centered at 1025 cm^−1^ is due to O-Si-O stretching, in which silicon atoms are located in tetrahedral coordination [42]. The bands recorded at 796 cm^−1^ and 694 cm^−1^ are assigned to vibrations in the silicon structure of the symmetric external Si-O bond [43,44]. The intense peak at 451 cm^−1^ corresponds to variations of the Si-O-H bond [45], while the peak of weak intensity at 525 cm^−1^ characterizes the presence of the Si-O-Si bond in diatomite [46]. A weak signal identified at 913 cm^−1^ corresponds to the vibration of Si-O-Al bonds and suggests the presence of clay in the diatomite [47].

The results of thermogravimetric analysis (TGA/DSC) (Figure 2) show a profound weight loss (9%) when natural diatomite was heated to 1000 °C. Other mass losses (5.1%) were detected within the range of 30–300 °C (DTG peak at 160 °C) and are due to loss of physically bound water (dehydration) from the diatomite surface [48]. The mass loss of 2.5% in the 290–500 temperature range (DTG peak at 350 °C) is due to the release of bound silanol groups (dehydroxylation) from the diatomite structure [49]. The obtained TGA/DSC results indicate that the observed peaks in the diatomite derivatogram correspond mainly to losses of water molecules of different chemical nature, starting from those adsorbed on the surface and those included in the internal structure as binding agents in the form of hydroxyl groups.

### 3.2. Characterization of the Synthesized (AgCl,Ag)NPs/Diatomite Composite

#### 3.2.1. Energy-Dispersive X-ray Spectroscopy (EDX) Studies

The results of SEM-EDX spectral analysis of natural diatomite and synthesized (AgCl, Ag)NPs/diatomite composite are shown in Figure 3A and Table 1. The results show a marked quantitative decrease in diatom of exchange cations such as Na, Mg, Al, K, Ca and Fe. The respective cations react with the anion, followed by the formation of soluble nitrate salts. The silver content in the prepared hybrid nanocomposites was found to be 0.71, 4.65 and 7.21% when using silver nitrate solution with initial silver concentrations of 100, 500 and 1000 mg/L, respectively (Table 1). 

The molar and mass correlations of silver and chlorine in the synthesized composites are presented in Table 2. These calculations show that the resulting silver-containing composites contain a mixture of AgCl-NPs and AgNPs and also indicate their ratios in the composites.

In the case of (AgCl,Ag)NPs/diatomite composite (Ag 0.71%), the ratio of AgCl-NPs:AgNPs nanoparticles is 98:2 (%), while in the case of (AgCl,Ag)NPs/diatomite composite (Ag 4.65%), the ratio is changed to 71:29 (%). In the case of the (AgCl, Ag)NPs/diatomite composite (Ag 7.21%), when a higher concentration of silver nitrate is used, the relative content of AgCl-NPs significantly decreases to 51:49 (%) (AgCl-NPs: AgNPs).

This phenomenon can be explained by the fact that the formation of AgCl-NPs in the synthesized composites is limited by the content of the halite mineral impurity in the diatomite structure.

#### 3.2.2. Powder X-ray Diffraction Analysis

X-ray patterns of natural diatomite and (AgCl, Ag)NPs/diatomite composites are shown in Figure 4. X-ray spectra confirm the presence of both reduced metallic silver nanoparticles and silver chloride nanoparticles in all the synthesized composites. The presence of metallic silver nanoparticles is indicated by the appearance of four peaks with 2θ values of 38.26°, 44.45°, 64.13° and 77.75° (ref. code: 00-004-0783) corresponding to reflections of the crystallographic planes (111), (200), (220) and (311). The emergence of six diffraction peaks with 2θ values of 27.96°, 32.38°, 46.37°, 54.96°, 57.59° and 76.84°, corresponding to reflections of the crystallographic planes (111), (200), (220), (311), (222) and (420), points to the presence of nanoparticles of chloride of silver (ref. code: 00-006-0480). The bright intensity of reflections is due to the higher content of AgCl-NPs compared with metallic silver nanoparticles, as well as the greater stability of crystal forms of silver chloride nanocrystallites.

#### 3.2.3. Transmission Electron Microscopy Studies

The transmission electron microscopy (TEM) results of all synthesized formulations are shown in Figure 5. TEM micrographs show the morphology, shape and dimensions of silver nanoparticles in synthesized (AgCl, Ag)NPs/diatomite hybrid composites with a concentration of immobilized silver equal to 4.65% and 7.21%. It can be observed that the spherical shapes of the nanoparticles are uniformly dispersed on the diatomite surface. Nanoparticles with sizes from 3 to 6 nm predominate. Nanoparticles found in the ranges of 8–10 nm and 1–2 nm are also present. Similar results were observed for (AgCl, Ag)NPs/diatomite composite containing 0.71% silver. The size of the nanoparticles decreases with decreasing initial silver concentrations in the solution used for the preparation of (AgCl, Ag)NPs/diatomite composites.

### 3.3. Antibacterial Activity

The minimal inhibitory concentrations (MIC) of (AgCl, Ag)NPs/diatomite composites on *Staphylococcus aureus* and *Klebsiella pneumonia* strains obtained in this study are presented in Table 3. According to Table 3, after 24 h of incubation, a similarity between the antimicrobial activity of 4.65% and 7.21% Ag/diatomite was observed in contrast to the 0.71% Ag/diatomite sample. Both 4.65% Ag/diatomite and 7.21% Ag/diatomite were more efficient than 0.71% Ag/diatomite and presented an inhibitory potential of about 2.5 mg/mL against *Staphylococcus aureus* and 5 mg/mL against *Klebsiella pneumonia*. Moreover, all the tested formulations showed low efficiency against *Klebsiella pneumonia*, but high inhibitory ability against *Staphylococcus aureus*. Furthermore, in the case of both bacterial strains, the same trend was noticed. However, all the tested samples presented a high antimicrobial effect against selected bacterial strains in comparison with the natural diatomite (control) where any inhibitory potential was recorded. This phenomenon is related to the immobilization of silver ions to the diatomite support and formation of (AgCl, Ag)NPs/diatomite formulations, which consequently causes the inhibitory effect. 

According to the literature data [10], the minimum inhibitory concentration of AgNPs/graphene oxide/diatomite composite against *E. coli* and *S. aureus* was found to be 5 and 10 mg/mL. A higher range (10.0–60.0 mg/mL) of antimicrobial effects has been shown for AgNPs/zeolite composites against *Streptococcus mutans*, *Lactobacillus casei*, *Candida albicans* and *Staphylococcus aureus* [17]. Another research group has presented the effect of nanosilver/diatomite composites in water treatment with a concentration of 5 mg/mL; the respective formulation inhibits almost 100% of *E. coli* cells [11]. In our research, the MIC values, based on the silver content in the hybrid composites, were found to be significantly lower in contrast to literature data (below 5 mg/mL). Moreover, the same inhibitory effect of (AgCl, Ag)NPs/diatomite formulations containing different concentrations of silver ions (4.65% and 7.21%) was observed. This does not exclude the fact that this phenomenon is related to the formation of different ratios between AgCl-NPs and AgNPs nanoparticles described in Section 3.2.1. The ratio of AgCl-NPs:AgNPs in composites is changed depending on the silver concentrations used in the composite synthesis. The AgNP content in the hybrid formulation increases with increasing silver concentrations. It is noteworthy that 4.65% (AgCl, Ag)NPs/diatomite contained a lower concentration of AgNPs than 7.21% (AgCl, Ag)NPs/diatomite formulations but showed the same inhibitory potential. This fact could indicate that AgCl-NPs have a greater inhibitory effect than AgNPs since (AgCl, Ag)NPs/diatomite formulations presented the same effect. This phenomenon can also be correlated with the Trojan horse action mechanism of (AgCl, Ag)NPs/diatomite composites, where the content of secondary oxidized silver ions on the surface of nanoparticles, e.g., AgCl-NPs/AgNPs, determines the antimicrobial effects [50]. The same phenomenon was noticed by [21] when tested Ag/mesoporous silica and AgCl/mesoporous silica composites exhibited similar antibacterial activity, but the concentration of AgCl/mesoporous silica nanocomposites being two times lower.

## 4. Conclusions

A method of synthesis of novel hybrid (AgCl, Ag)NPs/diatomite composites based on natural diatomite with a mineral impurity of halite was developed. A precursor of silver nanoparticles, silver nitrate solution, was used in different concentrations. One part of silver ions is converted into nanoparticles of silver chloride due to the exchange reaction of the silver nitrate solution with halite, while the other part of the silver ions is transformed into nanoparticles of metallic silver by means of a hydrogen peroxide agent. The presence of metallic silver nanoparticles and silver chloride nanoparticles in the synthesized composite is confirmed by X-ray structural analysis. The TEM analysis showed that the silver nanoparticles are uniformly distributed on the surface of the mineral matrix, and their sizes range from 1 nm to 10 nm with predominant sizes from 3 nm to 6 nm. Synthesized hybrid (AgCl, Ag)NPs/diatomite composites exhibit high antibiotic activity against Gram-positive and Gram-negative bacteria, indicating the prospect of their biomedical use as an antimicrobial agent. Synthesized hybrid (AgCl, Ag)NPs/diatomite composites with a dominant content of AgCl-NPs exhibit higher antibiotic activity. The hybrid (AgCl, Ag)NPs/diatomite composites possess antimicrobial potential. However, widespread use requires additional cytotoxic studies on eukaryotic systems, e.g., cell lines and animal models.

## Figures and Tables

**Figure 1 materials-13-03409-f001:**
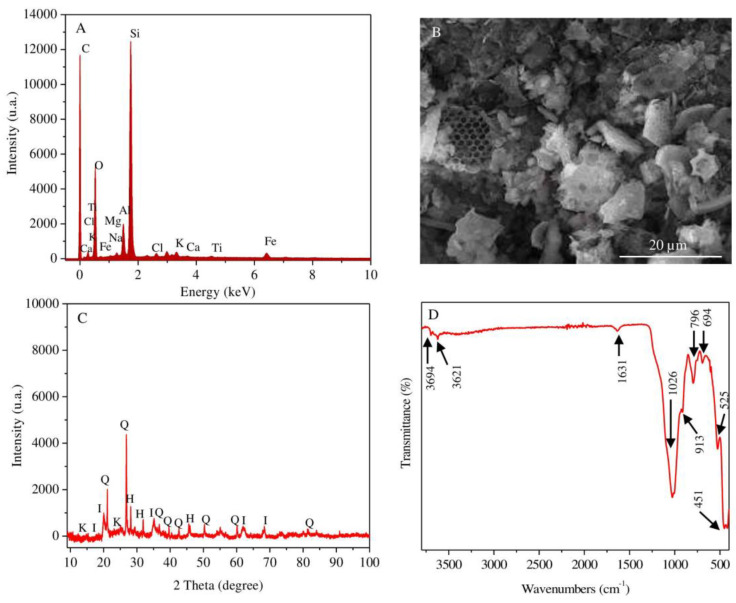
SEM-EDX spectrum (**A**) and SEM image (**B**); XRD patterns (I—illite; Q—quartz; K—kaolinite; H—halite) (**C**); FTIR spectrum; (**D**) curves of the studied raw diatomite samples.

**Figure 2 materials-13-03409-f002:**
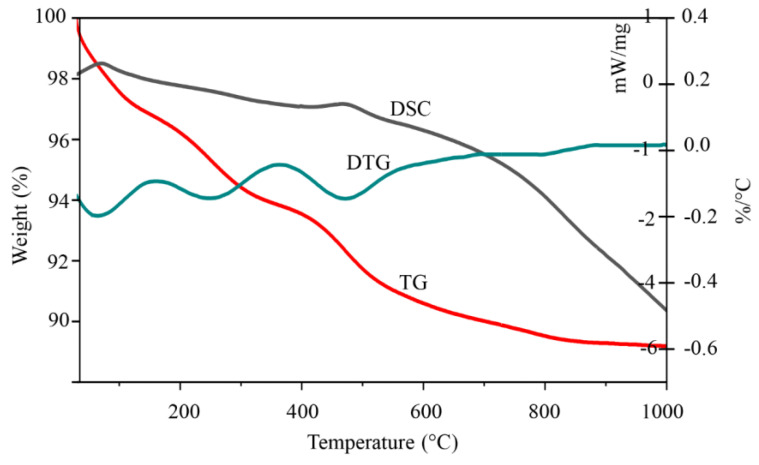
TGA curves of the studied raw diatomite sample.

**Figure 3 materials-13-03409-f003:**
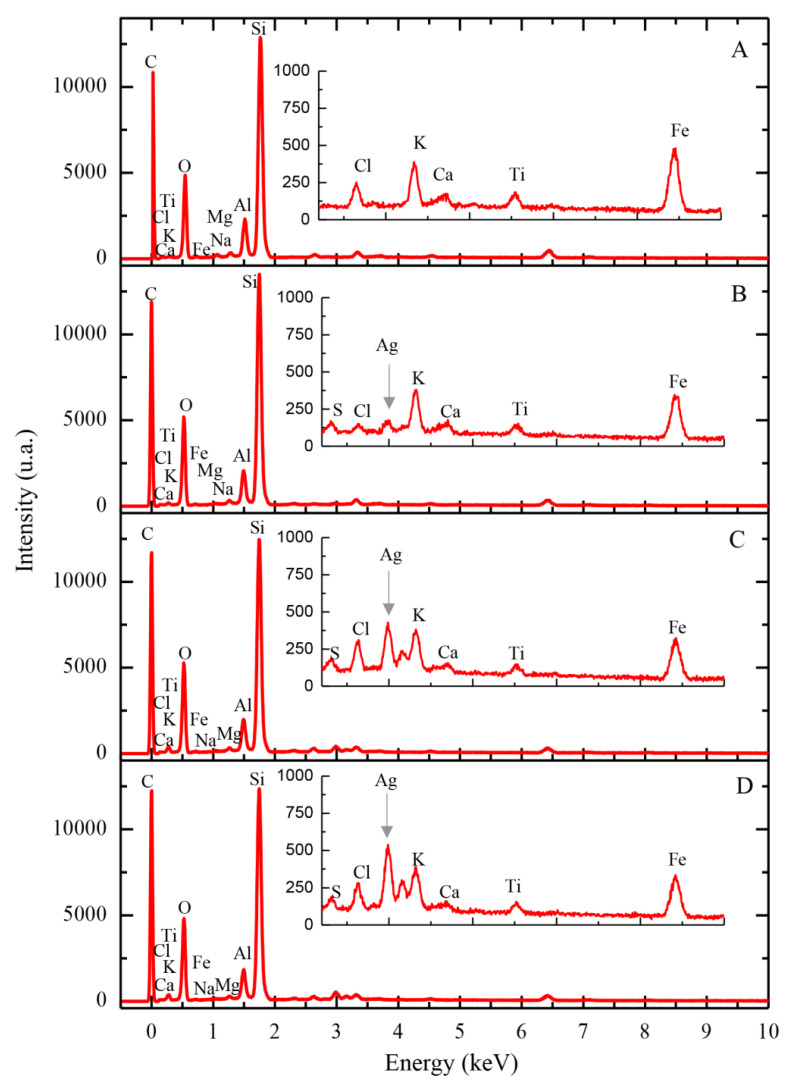
SEM-EDX spectra of the studied diatomite samples: (**A**) raw diatomite, (**B**–**D**) diatomite with immobilized silver (0.71, 4.65 and 7.21%, respectively).

**Figure 4 materials-13-03409-f004:**
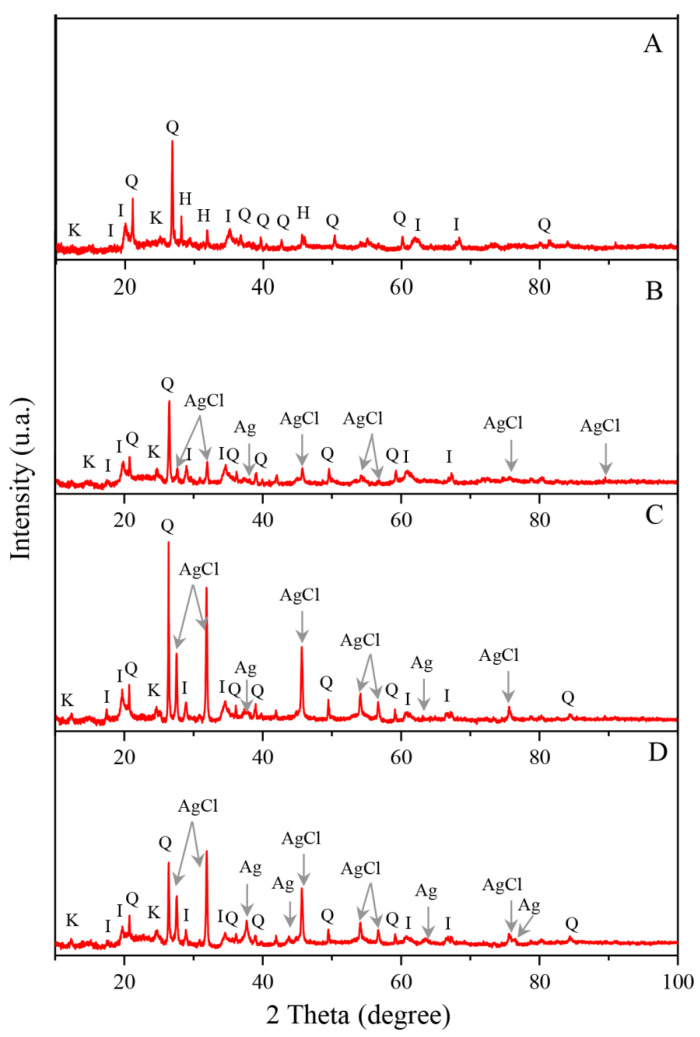
X-ray diffraction patterns of the studied diatomite samples: (**A**) raw diatomite, (**B**–**D**) diatomite with immobilized silver 0.71, 4.65 and 7.21%, respectively. I—illite, Q—quartz, K—kaolinite, H—halite, AgCl—silver chloride, Ag—metallic silver.

**Figure 5 materials-13-03409-f005:**
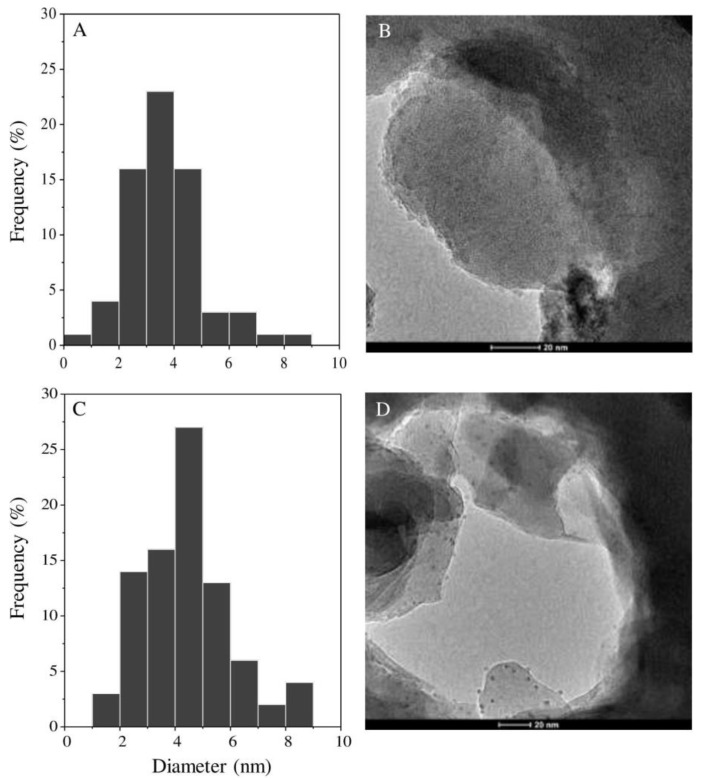
TEM micrographs of synthesized hybrid (AgCl, Ag)NPs/diatomite composites with silver concentrations of 4.65% (**A**,**B**) and 7.21% (**C**,**D**).

**Table 1 materials-13-03409-t001:** Elemental composition of natural diatomite and (AgCl, Ag)NPs/diatomite composite.

	C	O	Na	Mg	Al	Si	S	Cl	K	Ca	Ti	Fe	Ag
Natural diatomite	0.48	45.99	1.27	1.09	7.09	38.37	-	0.59	1.17	0.30	0.52	3.14	-
0.71% Ag/diatomite	0.80	50.31	0.72	0.87	5.64	35.76	0.43	0.23	1.18	0.30	0.40	2.65	0.71
4.65% Ag/diatomite	0.67	48.77	0.75	1.08	6.50	31.93	0.26	1.08	1.12	-	0.43	2.75	4.65
7.21%Ag/diatomite	0.66	49.37	0.73	0.94	5.75	30.05	0.22	1.21	1.04	-	0.49	2.34	7.21

**Table 2 materials-13-03409-t002:** Mass fraction and number of moles of silver, chlorine, silver chloride nanoparticles and metallic silver nanoparticles in the obtained composites.

	Ag		Cl		AgCl-NPs		AgNPs		AgCl-NPs:AgNPs
Samples	Mass%	n, mol	Mass%	n, mol	n, mol	%	n, mol	%	%
0.71% Ag/diatomite	0.71	0.066	0.23	0.065	0.065	98.4	0.001	1.5	98:2
4.65% Ag/diatomite	4.65	0.431	1.08	0.305	0.305	70.7	0.126	29.3	71:29
7.21% Ag/diatomite	7.21	0.668	1.21	0.341	0.341	51.1	0.327	48.9	51:49

**Table 3 materials-13-03409-t003:** Minimal inhibitory concentrations of (AgCl, Ag)NPs/diatomite composites.

*Bacteria Stains*	Minimum Inhibitory Concentration (MIC) of (AgCl, Ag)NPs/Diatomite Composite [mg/mL]			
	Natural Diatomite	0.71% Ag/Diatomite	4.65% Ag/Diatomite	7.21% Ag/Diatomite
*Staphylococcus aureus*	-	5 (0.036 Ag mg/mL)	2.5 (0.116 Ag mg/mL)	2.5 (0.180 Ag mg/mL)
*Klebsiella pneumoniae*	-	10 (0.071 Ag mg/mL)	5 (0.232 Ag mg/mL)	5 (0.360 Ag mg/mL)

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
