# Peer review of "Synthesis and Antibacterial Activity of (AgCl, Ag)NPs/Diatomite Hybrid Composite"

_materials, 2020, doi:10.3390/ma13153409_

Round 1
Reviewer 1 Report
This manuscript by Kubasheva et al. aims to assess several formulations of NPs diatomites for their antibacterial activity.
This manuscript deserves for a deep grammatical correction as there are several sentences that are very hard to understand. Then, in the introduction and the conclusion, the author did not contextualize their work in a more important context. What the challenge and issue associated with such composite materials? It must be clearly indicated rather than long sentences on the synthesis of such compounds. In fact only one sentence in introduction and one in conclusion focused on antibacterial activity, it is definitely not enough. Based on those limitations, and further comments, i recomment to reconsider this manuscript after major modifications. Specific comments are given below,
- "NP" you use this abbreviation everywhere without explain it, yet you fully write "nanoparticles" several times in the manuscript, please be homogeneous and use the abbreviation.(L15-L16 for example)
- In the abstract you speak about the used devices in the characterization of the samples, however, you must give some examples of the major results of your work too. Please rewrite the last part of the abstract.
- L30 Is diatomite a clay mineral? I'm not at all convinced about that, and hence, i d'ont understand the constant reminder with clay minerals
- L32-34 This sentence must be rewritten as in its current form it is impossible to understand.
- L38 "was" -> were
- L48 the sentence « Has also been », please rewrite
- L69 here you speak about “bioactive biopolymer” can you precise the link with your materials?
- L95 “the weight of used diatomite” or the weight of used Ag?
- L96 “we assumed” is there any reference to support this hypothesis?
- L112 “effectiveness”.. on what?
- L123 “natural diatomite served as a control” this is understandable, so why did you extensively discuss the characterization of diatomite?
- L124 “in triplicates” ok, however in the manuscript, the triplicated results, or standard error are never presented, please explain why?
- “Results and discussion” here I don’t understand the usefulness to distinguish natural and modified diatomite in two separated section. Based on the fact that diatomite is not investigated for antimicrobial properties, I suggest to merge these two sections. Another point, there is not any discussion in this section, please write a proper discussion after the presentation of your results.
- L128 please revise “while…while”
- L132-133 this sentence is unclear, please precise.
- L134 “the main minerals” you did not perform any rietveld refinement, what did you mean by “main” minerals?
- L147-148 “suggests the presence of clay” I think that XRD is more convincing on this point? The FTIR signal is very weak
- Fig2 at least you must label each curves, currently it is very hard to follow!
- Fig3 caption please correct the symbol into bracket
- I don’t understand why you label the samples according to the final Ag content, the results presented in Table 1 are therefore weird. Why not 1, 5 and 10%? Also, you should discuss somewhere the yield of Ag content. Obviously, this yield is higher at 5%, why?
- Table 2, please avoid any Cyrillic character in this table
- L185 “AgCl nanoparticles” what is it? From the beginning you speak about AgCl or AgNPs? Please clarify
- Fig 4 the relative intensity of Ag is weaker at 7.21% than at 4.65%?
- L244-245 the content of AgCl is also higher in 7.21%, so how can you explain the same inhibitory potential, please discuss!
- Table 3 Why is the Ag concentration different between the two bacteria strain, maybe I miss something?
- The conclusion is very descriptive and completely misses the target. You must contextualize your work correctly. Please rewrite.
Author Response
Response to Reviewer #1:
Thank very much you for your willingness to review our manuscript and valuable comments
This manuscript by Kubasheva et al. aims to assess several formulations of NPs diatomites for their antibacterial activity.
This manuscript deserves for a deep grammatical correction as there are several sentences that are very hard to understand. Then, in the introduction and the conclusion, the author did not contextualize their work in a more important context. What the challenge and issue associated with such composite materials? It must be clearly indicated rather than long sentences on the synthesis of such compounds. In fact only one sentence in introduction and one in conclusion focused on antibacterial activity, it is definitely not enough. Based on those limitations, and further comments, i recomment to reconsider this manuscript after major modifications. Specific comments are given below,
Re: Thank very much you for careful reading, and constructive suggestions for our manuscript that will help us to improve our work. According to your comments we have comprehensively revised our manuscript. Below we included the point-to-point response to each of your comments.
Point 1: "NP" you use this abbreviation everywhere without explain it, yet you fully write "nanoparticles" several times in the manuscript, please be homogeneous and use the abbreviation.(L15-L16 for example)
Re: According to the your suggestion, the term “NP” has been explained and the required corrections to the manuscript have been provided.
Point 2: In the abstract you speak about the used devices in the characterization of the samples, however, you must give some examples of the major results of your work too. Please rewrite the last part of the abstract.
Re: The last part of abstract has been rewritten as it was suggested.
Point 2: L30 Is diatomite a clay mineral? I'm not at all convinced about that, and hence, i d'ont understand the constant reminder with clay minerals
Re: Of cause, diatomite clearly does not belong to the group of clay minerals. In this case we pointed only that clay minerals are most often used as metal nanoparticles carriers. This sentence has been corrected.
Point 3: L32-34 This sentence must be rewritten as in its current form it is impossible to understand.
Re: This sentence has been rewritten.
Point 4: L38 "was" -> were
Re: The line 38 has been improved by the changing of the term “was” to “were”.
Point 5: L48 the sentence « Has also been », please rewrite
Re: This sentence has been corrected
Point 6: L69 here you speak about “bioactive biopolymer” can you precise the link with your materials?
Re: This sentence has been corrected: “In perspective, as an application, the obtained formulations can be found as bioactive and biocompatible materials in medicine, cosmetic or food industry (packaging materials)”.
Point 7: L95 “the weight of used diatomite” or the weight of used Ag?
Re: We would like to thanks for this remark. In this case it mean the weight of used diatomite. This sentence has been corrected: “A natural diatomite mass of 1.0 g was added to a silver nitrate solution (100 mL) with a following Ag ions concentration: 100 mg/L, 500 mg/L and 1000 mg/L in calculation 1, 5 and 10% of silver to mass of used diatomite”.
Point 8: L96 “we assumed” is there any reference to support this hypothesis?
Re: We have been rewritten this part of text and added reference.
Point 9: L112 “effectiveness”.. on what?
Re: Authors mean about effectiveness on antimicrobial properties of synthesized composites. Taken into consideration this remark, the text has been rewritten.
Point 10: L123 “natural diatomite served as a control” this is understandable, so why did you extensively discuss the characterization of diatomite?
Re: Taking into consideration this remark, the text has been rewritten. By using expression “natural diatomite served as a control” the authors mean the raw material (natural diatomite) without modification by silver nanooparticles has been used as a control.
Point 11: L124 “in triplicates” ok, however in the manuscript, the triplicated results, or standard error are never presented, please explain why?
Re: In this study the synthesis of the hybrid (AgCl, Ag)NPs/diatomite) composites have been performed in triplicates. But it is true, in this case only one series of the obtained composites was analyzed using instrumental methods. Therefore the standard error are not presented. The text in this section has been corrected.
Whereas the experiments of the antibacterial activity have been performed in triplicates. But presentation of the standard errors is also impossible according to the specifics of the used method. Minimal inhibitory concentration assay based on dilution methodology, according to CLSI standards and Resazurin-based 96-well plate microdilution method. The MIC value was determined by the changes of the indicator color of Resazurin from blue to pink. Considered this remark the section 2.4 has been supplemented with the missed information.
Point 12: “Results and discussion” here I don’t understand the usefulness to distinguish natural and modified diatomite in two separated section. Based on the fact that diatomite is not investigated for antimicrobial properties, I suggest to merge these two sections. Another point, there is not any discussion in this section, please write a proper discussion after the presentation of your results.
Re: In our situation, the research results have been obtained using a number of different instrumental methods. We fear that combining of the discussions of all the results obtained into a separate chapter will be difficult and perhaps not beneficial for the reader. So we will be very grateful if you don't argue to keep the current structure of the "Results and discussion" chapter.
Point 13: L128 please revise “while…while”
Re: This sentence has been improved.
Point 14: L132-133 this sentence is unclear, please precise.
Re: Taking into consideration remark, the sentence has been rewritten.
Point 15: L134 “the main minerals” you did not perform any riveted refinement, what did you mean by “main” minerals?
Re: We would like to thanks for this remark. In this case it was means the main minerals which have been detected in mineral composition of diatomite. Diatomite is siliceous sedimentary rock (not monomineral) and can be consist amorphous silica (fossilized frustules of diatoms) and quartz in quantities 60-90%, clay minerals, iron oxides and another little mineral impurities. In this context, the text has been corrected.
Point 16: L147-148 “suggests the presence of clay” I think that XRD is more convincing on this point? The FTIR signal is very weak.
Of course you are right, XRD is the primary method in the identification of minerals (XRD analysis results of this study are given in text above). FTIR analysis was performed for characterization of functional groups of natural diatomite. Nevertheless, the signal revealed at 913 cm−1 (Si-O-Al bonds) on the obtained FTIR spectrum additionally indicate the presence of aluminosilicates, especially clay minerals, in the diatomite.
Point 17: Fig 2 at least you must label each curves, currently it is very hard to follow!
Re: Thanks for the insightful remark – The Figure 2 has been improved with the messed information.
Point 18: Fig3 caption please correct the symbol into bracket
Re: Thank you for the valuable and constructive remark. The respective caption has been corrected.
Point 19: I don’t understand why you label the samples according to the final Ag content, the results presented in Table 1 are therefore weird. Why not 1, 5 and 10%? Also, you should discuss somewhere the yield of Ag content. Obviously, this yield is higher at 5%, why?
Re: Thanks for noticing an important aspect of our work related the content of Ag. We have included in Table 1 the real silver contents detected in the synthesized hybrid composites because 1, 5 and 10% are only the predicted silver contents, assuming that 100% of silver will be immobilized on the diatomite. In our case, the silver immobilization efficiency was 71.0, 92.5 and 72.1% respectively.
Point 20: Table 2, please avoid any Cyrillic character in this table
Re: The authors apologies for this mistake. The corrections regarding the Cyrillic character in the Table 2 have been performed.
Point 21: L185 “AgCl nanoparticles” what is it? From the beginning you speak about AgCl or AgNPs? Please clarify
Re: We are grateful for such thorough analysis of our results. So, the main aim of study was synthesis of the hybrid (AgCl, Ag)NPs/diatomite) composite based on a natural diatomite containing silver nanoparticles in the form of metallic silver (AgNPs) and silver chlorides (AgCl-NPs). The presence of metallic silver nanoparticles in the forms of metallic silver and silver chlorides was confirmed by XRD analysis
Point 22: Fig 4 the relative intensity of Ag is weaker at 7.21% than at 4.65%?
Re: We are very grateful for a thorough look at respective part of the work. Yes, the authors are agree with you remark and the intensity of Ag signal it can be results from the surface saturation of diatomite. It can also be seen that the silver peaks are more intense than that of quartz peaks for 7.21% Ag diatomite then for 4.65% Ag diatomite.
Point 23: L244-245 the content of AgCl is also higher in 7.21%, so how can you explain the same inhibitory potential, please discuss!
Re: We would like to thanks for the relevant comment. The content of silver chloride nanoparticles in the (AgCl, Ag)NPs/diatomite composite is limited by the content of the halite in the diatomite (please see also table 1). Therefore, the content of AgCl-NPs is not higher in 7.21% Ag diatomite, so it can to explain the same inhibitory potential.
According to this remark the text has been improved.
Point 24: Table 3 Why is the Ag concentration different between the two bacteria strain, maybe I miss something?
Re: Thank you for valuable remark. The data presented in table 3 regarding the Ag concertation results from the determined values of minimal inhibitory concentration and total silver content in AgCl,AgNPs/diatomite composites per applied samples.
Point 25: The conclusion is very descriptive and completely misses the target. You must contextualize your work correctly. Please rewrite.
Re: Thank you for constructive remark - the conclusion section has been rewritten in order to follow the target.
Reviewer 2 Report
It is interesting to use natural diatomite to produce AgCl NPs/Diatomite composite. The description about the characterizations of hybrid composite is quite comprehensive and clear. However, the Table 3 is quite confusing and requires modification. There are some errors on word spelling and format requiring correction.
The information provided in the introduction is not sufficient to justify the reason for using AgCl NPs/Diatomite composite. In addition, it would be better to provide some explanation or hypothesis about the lower antibacterial activity of the AgCl NPs/Diatomite composite used in this study.
Author Response
Response to Reviewer #3:
Thank very much you for your willingness to review our manuscript and valuable comments
Point 1: It is interesting to use natural diatomite to produce AgCl NPs/Diatomite composite. The description about the characterizations of hybrid composite is quite comprehensive and clear. However, the Table 3 is quite confusing and requires modification. There are some errors on word spelling and format requiring correction.
Re: We are grateful for such thorough analysis of our results and therefore the enumerated comments and suggestion have been taken in full consideration and the required corrections have been performed.
Point 2: The information provided in the introduction is not sufficient to justify the reason for using AgCl NPs/Diatomite composite. In addition, it would be better to provide some explanation or hypothesis about the lower antibacterial activity of the AgCl NPs/Diatomite composite used in this study.
Re: We would like to thanks for the relevant comment. According to the Reviewer’s remarks:
- The introduction has been improved with the missed information regarding the reason of using (AgCl, Ag) NPs/diatomite composite.
- Regarding the lower antimicrobial activity of the obtained hybrid (AgCl Ag)NPs/diatomite composite in present study - This phenomenon, it can be also correlated with Trojan horse action mechanism of (AgCl,Ag) NPs/diatomite composites, where content of secondary oxidized silver ions for surface of nanoparticles eg. AgCl/Ag NPs determine the antimicrobial effects. According to this remark the text has been improved with the messed information.
Reviewer 3 Report
Dear Authors,
After the review process, I have several comments:
you should clearly present the aim of the paper in the abstract section;
you should expand the introduction section, it is too short and provides only general data. They should insert data related to this subject published in mdpi, for example https://doi.org/10.3390/nu10050607;
you should insert a statistical section at the end of Materials and Methods;
you should insert references al all Materials and Methods sections;
you should present the significance of the results and comment the possible limitation of the study;
you should explain why they used only these two bacterial strains.
Best regards!
Author Response
Response to Reviewer #2:
Thank very much you for your willingness to review our manuscript and valuable comments
After the review process, I have several comments:
Point 1: you should clearly present the aim of the paper in the abstract section;
Re: Thank you for valuable suggestion, therefore the Reviewer’s comment has been taken in consideration and the abstract has been revised and rewrite.
Point 2: you should expand the introduction section, it is too short and provides only general data. They should insert data related to this subject published in mdpi, for example https://doi.org/10.3390/nu10050607;
Re: Thank you for constructive remark. The reference has been added to the revised version of introduction.
Point 3: you should insert a statistical section at the end of Materials and Methods;
Re: We would like to thanks for the relevant comment. Uunfortunately, our work does not contain such data to which statistical processing methods could be applied. Generally, these are the data obtained using instrumental methods. An explanation regarding the presentation of the standard deviation values is also given in the responses to the Reviewer 1.
Point 4: you should insert references al all Materials and Methods sections;
Re: The additional references have been added to the M&M section of revised manuscript.
Point 5: you should present the significance of the results and comment the possible limitation of the study;
Re: The obtained hybrid (AgCl, Ag) NPs/diatomite composites possess the antimicrobial potential, however the widespread use require additional cytotoxic studies on eukaryotic systems e.g. cell lines, animal model. The additional information have been added to the conclusion section.
Point 6: you should explain why they used only these two bacterial strains.
Re: We are very grateful for a thorough look at respective part of the work related to the antimicrobial activity of the obtained (AgCl, Ag) NPs/diatomite composites. In the present research have been used the most common microorganisms in medical field as a drug resistance microorganisms. For this purpose have been chosen one Gram-positive (Staphylococcus aureus) and one Gram-negative (Klebsiella Pneumoniae) bacteria in order to check any effectiveness of the synthesized nanocomposites. However the widespread use require additional studies which is of course, of our interest for further perspective.
Round 2
Reviewer 1 Report
The authors made the required modifications ont their manuscript based on the reviewer' recommendations. The manuscript is now suitable for publication in Materials.
Reviewer 3 Report
Dear Authors,
I do not have other comments.
Best regards!